# Differential Expression of Lonp1 Isoforms in Cancer Cells

**DOI:** 10.3390/cells11233940

**Published:** 2022-12-06

**Authors:** Giada Zanini, Valentina Selleri, Anna De Gaetano, Lara Gibellini, Mara Malerba, Anna Vittoria Mattioli, Milena Nasi, Nadezda Apostolova, Marcello Pinti

**Affiliations:** 1Department of Life Sciences, University of Modena and Reggio Emilia, 41125 Modena, Italy; 2National Institute for Cardiovascular Research (INRC), 40126 Bologna, Italy; 3Department of Medical and Surgical Sciences for Children and Adults, University of Modena and Reggio Emilia, 41124 Modena, Italy; 4Department of Surgical, Medical, Dental and Morphological Sciences, University of Modena and Reggio Emilia, 41124 Modena, Italy; 5Department of Pharmacology, University of Valencia, 46010 Valencia, Spain; 6FISABIO—Hospital Universitario Dr. Peset, 46017 Valencia, Spain

**Keywords:** Lon protease, mitochondria, SW620, mitochondrial DNA

## Abstract

Lonp1 is a mitochondrial protease that degrades oxidized and damaged proteins, assists protein folding, and contributes to the maintenance of mitochondrial DNA. A higher expression of LonP1 has been associated with higher tumour aggressiveness. Besides the full-length isoform (ISO1), we identified two other isoforms of Lonp1 in humans, resulting from alternative splicing: Isoform-2 (ISO2) lacking aa 42-105 and isoform-3 (ISO3) lacking aa 1-196. An inspection of the public database TSVdb showed that ISO1 was upregulated in lung, bladder, prostate, and breast cancer, ISO2 in all the cancers analysed (including rectum, colon, cervical, bladder, prostate, breast, head, and neck), ISO3 did not show significant changes between cancer and normal tissue. We overexpressed ISO1, ISO2, and ISO3 in SW620 cells and found that the ISO1 isoform was exclusively mitochondrial, ISO2 was present in the organelle and in the cytoplasm, and ISO3 was exclusively cytoplasmatic. The overexpression of ISO1 and, at a letter extent, of ISO2 enhanced basal, ATP-linked, and maximal respiration without altering the mitochondria number or network, mtDNA amount. or mitochondrial dynamics. A higher extracellular acidification rate was observed in ISO1 and ISO2, overexpressing cells, suggesting an increase in glycolysis. Cells overexpressing the different isoforms did not show a difference in the proliferation rate but showed a great increase in anchorage-independent growth. ISO1 and ISO2, but not ISO3, determined an upregulation of EMT-related proteins, which appeared unrelated to higher mitochondrial ROS production, nor due to the activation of the MEK ERK pathway, but rather to global metabolic reprogramming of cells.

## 1. Introduction

Alternative splicing is a tightly regulated process allowing for the expression of multiple RNA and protein isoforms from one gene, and its dysregulation has recently been described in several diseases, most of them due to either the alteration in spliceosomal/splicing regulatory factors expression or genetic mutation in canonical RNA splicing sites [1,2]. Recurrent somatic mutations in components of the human splicing machinery have occurred in neurological diseases, such as Parkinson’s disease [3,4] and Alzheimer’s disease [5], infectious and immunological diseases, such as cardiovascular disease [6,7], systemic lupus erythematosus [8], diabetes mellitus [9,10], viral infections [11], and inflammatory response [12], solid tumours, and hematological malignancies [1]. A dysregulated splicing program in cancer can represent a driving force in the development and maintenance of cancer [13]. For instance, tumour suppressor genes that undergo alternative splicing in cancer, such as p53, can lose partially or totally their function and generate several isoforms which are able to maintain the abnormal proliferative and apoptotic rhythm [14,15,16]. Moreover, alternative splicing events can affect the cellular response to nutrient deprivation and hypoxia, which are common characteristics of many solid tumours [17], and angiogenesis, which is a major characteristic of cancer maintenance and progression [17,18]. Furthermore, the association between alternative splicing events and immune systems also affects the occurrence and development of cancer [17,19]. 

The mitochondrial protease Lonp1 is a multifunctional enzyme present in the mitochondrial matrix, where it degrades oxidized or damaged proteins, contributes to the proper folding of proteins imported into the organelle, and binds mitochondrial DNA (mtDNA). Through these multiple functions, Lonp1 plays a crucial role in several cellular functions, including respiration, synthesis of steroids, and mitophagy [20,21,22]. 

Because of its crucial role in mitochondrial homeostasis, Lonp1 can also contribute to neoplastic transformation and cancer progression [22]. Lonp1 plays a key role in the metabolic reprogramming of cancer cells by remodelling OXPHOS complexes and protecting against senescence. Accordingly, Lonp1 is upregulated in several types of cancer, and its higher expression has been associated with higher tumour aggressiveness both in vitro and in vivo [21,23]. The overexpression of Lonp1 has been shown to promote cell proliferation, apoptotic resistance to stresses, and transformation. At least in some cell models, Lonp1 overexpression can drive epithelial to mesenchymal transition (EMT) by increasing mitochondrial reactive oxygen species (ROS) production, which in turn activates ERK and p38 MAPK pathways [23]. An enhanced protease activity of Lonp1, caused by Akt-mediated phosphorylation, contributes to primary and metastatic tumour growth in vivo [24]. 

Recently, we and others have shown that Lonp1 can also have some extra-mitochondrial functions [25,26]; we detected Lonp1 in the nucleus, where as much as 22% of all cellular Lonp1 can be found, as well as in the mitochondria-associated membranes (MAMs). Nuclear localization is detectable under all conditions, but the amount is dependent on proteostress response, such as the response to heat shock (HS), while Lonp1 is present in the MAMs under stress conditions. These observations suggest that extramitochondrial Lonp1 can contribute to the survival of cancer cells in response to stress conditions frequently present in the tumour microenvironment. 

Here, we report that the presence of Lonp1 outside the mitochondria is due to the coexistence of three different Lonp1 isoforms generated by alternative splicing. These three Lonp1 isoforms localize in different cell compartments, as the full-length, main isoform is expressed almost exclusively in the mitochondria. We also show that their expression is highly variable in cancer cells of different origins, suggesting a different biological role for Lonp1 isoforms in cancer development and progression. 

## 2. Materials and Methods

### 2.1. Cell Culture and Transfections

SW480 and SW620 were maintained in Glutamax RPMI 1640 supplemented with gentamycin and 10% fetal bovine serum (FBS, Life Technologies, Carlsbad, CA, USA). SW480 and SW620 cell lines were kindly gifted by Dr. Zanocco Marani. HeLa cells were maintained in Glutamax Dulbecco modified Eagle medium (DMEM) supplemented with gentamycin and 10% FBS. HMC3 cells were maintained in Eagle’s minimum essential medium (EMEM) supplemented with 6 g/L glucose, 10% of fetal calf serum (FCS), 2 mM glutamine, 100 μg/mL streptomycin, and 100 U/mL penicillin. HepG2 cells were maintained, and DMEM was supplemented with 10% FBS. The A594 cell line was maintained in the DMEM/F12 medium supplemented with 10% FBS. A375 cells were maintained in DMEM supplemented with 10% FBS. Hela, HMC3, HepG2, A594, and A375 cells were obtained from the American Type Culture Collection (ATCC). Skin fibroblasts were maintained in DMEM with the addition of 10% FBS, 100 μg/mL streptomycin, and 100 U/mL penicillin, and dental pulp stem cells (PDSC) cells were maintained in DMEM supplemented with 20% FBS and penicillin 500 U/mL, streptomycin 500 µg/mL, amphotericin B 1.25 µg/mL. The lymphocytes were maintained in Glutamax RPMI 1640 containing 10% FBS, 1% penicillin/streptomycin, and 1 μg/mL phytohemagglutinin (PHA). Fibroblasts, dental pulp stem cells (DPSC), and lymphocytes were derived from healthy donors after obtaining clearance from the local ethical committee and signed informed consent (Ethical committee approval: ref. number 184/10. released by Comitato Etico Emilia Romagna Area Vasta Nord). Cells were maintained in an incubator at 37 °C, 5% CO_2_, in a humidified atmosphere.

### 2.2. Cell Transfection and Retroviral Transduction

The pMSCV-Puro empty vector and the pMSCV containing the cDNA encoding for the Lonp1 isoform 1 (pMSCV-Lon-ISO1), isoform 2 (pMSCV-Lon-ISO2), and isoform 3 (pMSCV-Lon-ISO3) were used to transiently transfect the amphotrophic Phoenix (phxA) cells. Transient transfection was performed by using lipofectamine 3000 (Life Technologies Corporation) and 20 μg of pMSCV-Lon-ISO1/ISO2/ISO3 plus 0.9 μg of helper plasmid. Retroviral supernatants were used to stably transfect SW620 cells, which were selected and maintained in a medium supplemented with 2 μg/mL puromycin. Transient transfection was performed with Lipofectamine 3000 (Life Technologies) following the provider’s instructions. The transfection efficiency of SW620 cells and HeLa cells was always >40%; the cells were analysed three days post-transfection.

### 2.3. Western Blotting

Western blotting was performed as previously described [21]. The total cell lysates were prepared in a RIPA lysis buffer plus protease inhibitor cocktail (Sigma Aldrich, St. Louis, MO, USA) and phosphatase inhibitors (Sigma Aldrich). Nuclear, mitochondrial, and cytosolic fractions were isolated by using the cell fractionation kit (Abcam, Cambridge, UK), following the provided instructions. Samples were resolved by SDS-PAGE on precast gels (8%, 12%, 4–12%, ThermoFisher Scientific, Waltham, MA, USA) and transferred to nitrocellulose membranes (Bio-Rad Laboratories), which were then immunoblotted. The following primary antibodies were used: anti-Lonp1 (Abcam, Cambridge, UK, #ab103809), anti-LaminB1 (Abcam, Cambridge, UK, #ab16048), anti-β-actin (Abcam, Cambridge, UK, #ab8227), anti-TOM20 (Santa Cruz Biotechnology, Dallas, TX, USA), anti-GFP (Fine Biotech, Wuhan, China, #FNab03434), anti-LDHA (Cohesion Biosciences, London, UK, #CQA2055), anti-GLUT1 (Cohesion Biosciences, London, UK, #CPA2071), anti-Mfn1 (Cohesion Biosciences, London, UK, #CQA3646), anti-Mfn2 (Abgent, San Diego, CA, USA, #RB21136), anti-Opa1 (Assay Biotech, Fremont, CA, USA, #R12-2283), anti-DRP1 (ABclonal Technology, Woburn, MA, USA, #A2586), anti-Snail (ABclonal Technology, Woburn, MA, USA, #A5243), anti-Slug (Cohesion Biosciences, London, UK, #CPA3014), anti-Ecad (Cohesion Biosciences, London, UK, #CPA1198), anti-Ncad (Cohesion Biosciences, London, UK, #CPA1200), anti-p-ERK (Cell Signaling, Danvers, Massachusetts, USA, #9106), anti-p-MEK (Cell Signaling, Danvers, Massachusetts, USA, #9121), anti-p38 (Cell Signaling, Danvers, Massachusetts, USA, #9212), anti-Akt (Cell Signaling, Danvers, Massachusetts, USA, #9272), and anti-p-Akt (Cell Signaling, Danvers, Massachusetts, USA, #9271). The following secondary antibodies were used: HRP-conjugated goat anti-rabbit and HRP-conjugated goat anti-mouse (Bio-Rad Laboratories). The enhanced chemiluminescent (ECL) clarity chemiluminescent substrate (Bio-Rad Laboratories) was used to detect proteins by using a Chemidoc MP (Bio-Rad Laboratories). Image analysis was performed by Image Lab software v5.2.1.

### 2.4. Immunofluorescence and Confocal Microscopy

Cells for confocal microscopy were grown on coverslips, fixed with 3.7% formaldehyde (Sigma Aldrich) in PBS for 9 min, acetone for 5 min at -20°C, and permeabilized with 0.1% triton X-100 in PBS, for 6 min. Cells were blocked in 3% bovine serum albumin (BSA) for 30 min, incubated with primary antibodies anti-Lonp1 (Abcam, Cambridge, UK, #ab103809) and anti-Mitochondria (EMD Millipore Corp, USA, MAB1273) in 3% BSA for 1 h, washed, incubated with Alexa Fluor-conjugated secondary antibodies (Alexa Fluor^®^ 488 Goat anti-Mouse A-11017, Alexa Fluor^®^ 647 Goat anti-Rabbit A-21246, Life Technologies) in 3% BSA for 1 h, and washed again to remove the unbound antibody. Cells were counterstained with DAPI 0.5 ug/mL. Coverslips were mounted in Fluoromount (Sigma Aldrich), and images were collected on a confocal microscope SP8-AOBS (Leica). Image analysis was performed with ScanR. 

### 2.5. Mitochondrial Network Analysis

Mitochondrial network analysis was performed on 3D images, with 16 z-stacks per image, using the “Mitochondria analysis” plugin of Fiji (ImageJ) [27]. Briefly, mitochondria have been identified using the weighted mean thresholding method after the optimization of 3D threshold parameters. Then, the morphological (mitochondrial total number and sphericity) and network (mean branch diameter) parameters were automatically calculated. 

### 2.6. RNA Extraction and Quantification of Lonp1 Isoforms

The total RNA was extracted from cells by the Quick-RNA Miniprep Plus Kit (Zymo Research, Irvine, CA, USA) following the manufacturer’s instruction, and the amount of RNA was quantified using the NanoDrop ND-1000 (Thermo Fisher Scientific, Inc.). Then, 1 µg of RNA was reverse transcribed using the iScript cDNA synthesis kit from Bio-Rad, Hercules, CA, USA.

The amount of the different isoforms of LONP1 was quantified by droplet digital PCR (ddPCR). Two different reaction mixtures of ddPCR were prepared, one multiplex for isoforms 1 and 3 and another for isoform 2. The reaction mixture of ddPCR had this composition in 20 µL of the final volume: 10 ng of DNA samples, 2X ddPCR Supermix for Probes (Bio-Rad Laboratories), and 1,1 µL of each isoform custom assay (Bio-Rad Laboratories). The thermal protocol conditions were 95 °C for 10 min, followed by 40 cycles of 94 °C for 30 s and 58 °C for 1 min, then 98 °C for 10 min. Droplet reading was performed on QX200 ddPCR droplet reader (Bio-Rad laboratories). Analysis was performed using QuantaSoft Analysis software (version 1.7.4.0917, Bio-Rad laboratories). The following primers were used: LONP1 isoform 1 forward: 5′-CACTGCAGCAGGAGC-3′, LONP1 isoform 1 reverse: 5′-TGCCCAAGGAGGAGAG-3′, LONP1 isoform 1 probe: 5′-CGAGGCcAGCGGA-3′, LONP1 isoform 2 forward: 5′-CACTGCAGCAGGAGC-3′, LONP1 isoform 2 reverse: 5′-ATCGTCATGGGCGTG-3′, LONP1 isoform 2 probe: 5′-ATGACCGGGcCTCG-3′, LONP1 isoform 3 forward: 5′-AGCTCCCGGCTGAG-3′, LONP1 isoform 3 reverse: 5′-GTGACCTGGAAGTCCTC-3′, and LONP1 isoform 3 probe: 5′-TTCTCTACCTcCACCATGA-3′. The ISO2 primers target a joining region between exon 1a and 1b and are unique to ISO2. The ISO1 primers target the 5′ regions included exclusively in the ISO1 mRNA, while the ISO3 primers target all the isoforms. Thus, the number of copies of ISO3 can be indirectly calculated by subtracting the number of copies of ISO1 and ISO2 from the ISO3 number of copies detected. Moreover, the relative percentage of every single isoform mRNA has been calculated as the ratio between the absolute number of mRNA copies of the specific isoform and the absolute number of mRNA copies of all three isoforms multiplied by 100. 

### 2.7. DNA Extraction and mtDNA Quantification

Total DNA was extracted from cells by DNeasy Blood & Tissue Kit (QIAGEN, Hilden, Germany) following the manufacturer’s instruction, and the amount of DNA was quantified using the NanoDrop ND-1000 (Thermo Fisher Scientific, Inc.). Then, 10 µL of DNA were digested using FastDigest HindIII (Thermo Fisher Scientific, Inc.).

The amount of mtDNA was quantified by ddPCR. The reaction mixture of ddPCR had this composition in 20 µL of the final volume: 10 ng of DNA samples, 2X ddPCR Supermix for Probes (Bio-Rad Laboratories), and 1.1 µL of each isoform custom assay (Bio-Rad Laboratories). The thermal protocol conditions were: 95 °C for 10 min, followed by 40 cycles of 94 °C for 30 s, and 55 °C for 1 min, then 98 °C for 10 min. Droplet reading was performed on a QX200 ddPCR droplet reader (Bio-Rad laboratories). Analysis was performed using QuantaSoft Analysis software (version 1.7.4.0917, Bio-Rad laboratories). The following wet-validated primers were used: ND2 (Bio-Rad, Hercules, CA, USA, #10031252) and Actb (Bio-Rad, Hercules, CA, USA, #10042961). 

### 2.8. Mitochondrial Bioenergetics Assay

The real-time measurement of the oxygen consumption rate (OCR) was performed by an XFe96 Seahorse Extracellular Flux Analyzer (Agilent, Santa Clara, CA, USA) by using the MitoStress Kit (Agilent) according to the manufacturer’s procedures. A total of 80,000 cells/well were seeded, and the OCR was measured in XF media (non-buffered RPMI medium containing 10 mM glucose, 2 mM L-glutamine, and 1 mM sodium pyruvate) under basal conditions and in response to 2 μM oligomycin, 1.5 μM of carbonylcyanide-4-(trifluoromethoxy)-phenylhydrazone (FCCP), and 1 μM of Antimycin and Rotenone (all from Sigma Aldrich). Indices of the mitochondrial respiratory function were calculated from the OCR profile: basal OCR (before the addition of oligomycin), ATP-linked OCR (calculated as the difference between the basal OCR rate and oligomycin-induced OCR rate), maximal OCR (calculated as the difference of the FCCP rate and antimycin plus rotenone rate), spare respiration capacity (SRC, the difference between the basal ATP production and its maximal activity), and non-mitochondrial oxygen consumption. Each experiment was repeated three times, each in triplicate. 

### 2.9. Reactive Oxygen Species Measurement

Mitochondrial ROS (mtROS) levels were quantified by measuring the fluorescence MitoSOX Red (Thermo Fisher Scientific, Inc.) according to the manufacturer’s procedures. A total of 50,000 cells were seeded in a Lumux multiwell-96 (Sarstedt, Germany) and incubated with 5 µM MitoSOX Red for 10 min at 37 °C in the dark. Then, the cells were washed three times with 1X PBS, and cellular fluorescence was measured after 30 min using the Fluoroskan FL Microplate Fluorometer and Luminometer (Thermo Fisher Scientific, Inc.) with excitation λ 544 nm, emission λ 590 nm, and time exposure of one second. 

### 2.10. Soft Agar Colony Formation Assay

The soft agar colony formation assay was performed by seeding in 0.25% agarose top cells at a density of 50,000 cells/well in 6 well multiwell, above a 0.6% agarose base, and was incubated at 37 °C and 5% CO_2_, in a humidified atmosphere for three weeks. Then, cell colonies were left for 30 min with crystal violet 0.007% and counted using ImageJ. Experiments were performed in triplicate.

### 2.11. Statistical Analysis

All the measurement data are presented as the mean ± standard deviation (SD) if not otherwise specified. Statistical analysis between the different experimental conditions was performed with ANOVA followed by the Bonferroni means comparison or t-test when appropriate. An unpaired t-test was used to compare the levels of the different isoforms between the normal and tumour tissues in different cancer types. A threshold of *p* ≤ 0.05 was selected to indicate statistical significance. Statistical calculations were performed using the standard functions of GraphPad Prism 8.0. Confocal image analysis was performed by Fiji (ImageJ) v2.0 and ScanR with the plugins Mitochondria Analyzer [27]. 

## 3. Results

### 3.1. Lonp1 Is Present in Different Isoforms, Showing Differential Expression Pattern and Intracellular Distribution

The human Lonp1 isoform-1 protein (ISO1, NP_004784.2) is formed by 959 amino acids and has been highly conserved throughout evolution. However, an inspection of the public database revealed that at least two other isoforms are present in humans, resulting from the alternative splicing of exon 1 (Figure 1A and Appendix A). The Lonp1 Isoform-2 (ISO2, NP_001263408.1) 895-aminoacid is long and lacks amino acids 42-105, including a part of the mitochondrial targeting sequence (MTS); Lonp1 isoform-3 (ISO3, NP_001263409.1) contains 763 amino acids and lacks amino acids 1-196, including the entire MTS. The nuclear localization sequence (NLS) we previously identified [26] is present in all the isoforms. 

We evaluated the expression of the different Lonp1 isoforms in primary cells and cancer cell lines of different origins at transcriptional levels by ddPCR. The relative levels of ISO1, 2, and 3 are reported in Figure 1B. All the isoforms were detectable in any type of cell analysed, with a high degree of variability. ISO1 is the form expressed at the highest levels in most of the cell lines; the ISO2 levels are 5.6–7.8 folds less than ISO1. ISO3 is expressed at high levels in SW620 cells among cancer cell lines and in fibroblasts and DPSCs among primary cells.

Thus, we wondered if the expression of Lonp1 isoforms could be different in cancer cells if compared to normal, non-transformed cells. We interrogated the TSV database (http://tsvdb.com, accessed on 25 November 2019), a database reporting the relative levels of mRNA splicing variants concerning the expression of Lonp1 isoforms in different primary solid tumours compared with normal tissues (Figure 1C–E). We found that ISO1 was upregulated in lung, bladder, prostate, and breast cancer, while ISO2 was upregulated in all the other types of cancers we analysed, i.e., rectum, colon, cervical, bladder, prostate, breast, head, neck, and renal. Moreover, the expression of ISO3 was detectable in several cancer types, such as the rectum, cervical, bladder, breast, head, neck, and renal cancer, while its expression was not detectable in the corresponding normal cells, with the notable exception of head and neck epithelium. 

Furthermore, using TSVdb, we generated Kaplan–Meyer curves for all these primary solid tumours, stratifying patients on the basis of the expression of the three Lonp1 isoforms. We found that at least in some cases, the survival curves of patients with higher or lower ISO1 differed from those with higher or lower ISO2, or higher or lower ISO3 (Appendix A). This suggests, that besides the total expression level of Lonp1, the relative expression of the isoforms can impact the patients’ survival, as we can see, for example, in colon adenocarcinoma, head, and neck squamous cell carcinoma, and lung squamous cell carcinoma.

We have previously shown that Lonp1 can be present, besides mitochondria, in other cell compartments, namely the MAMs and the nucleus [25,26]. Thus, we hypothesized that these three isoforms could display different intracellular distributions and may localize independently. 

We, therefore, generated three constructs where each isoform was tagged at the C-term by eGFP or mCherry, and we analysed the intracellular distribution in the transfected SW620 cells. When cell lysates were probed with an anti-eGFP Ab, ISO1-eGFP, and ISO2-eGFP, the same MW of approximately 138 KDa was observed, which was slightly higher than predicted (Figure 2A, left panel) [28]. EGFP-ISO3 was lighter than ISO1-eGFP and ISO2-eGFP, with an MW of 112 KDa. The same samples were also probed with anti-Lonp1 Ab (Figure 2A, right panel). This Ab correctly detected the bands corresponding to the three Lonp1-eGFP isoforms but only evidenced the band corresponding to the endogenous ISO1 and ISO2 Lonp1, suggesting that the levels of ISO3 were normally low and could not be detected. Then, we again transfected the cells with eGFP-tagged isoforms and assayed cytosolic, mitochondrial, and nuclear fractions with an anti-eGFP antibody. The ISO1-eGFP was detected exclusively in the mitochondrial fraction (Figure 2B, upper panel), the ISO2-eGFP was present in the mitochondria and in the cytosol, and ISO3-eGFP was detected almost exclusively in the cytosol but not in the mitochondrial fraction (Figure 2B, lower panel). Consistently, confocal microscopy analysis showed that ISO1 was almost exclusively mitochondrial, ISO2 was mainly mitochondrial and partially cytosolic, while ISO3 was cytosolic but not mitochondrial, in agreement with the absence of the MTS (Figure 2C). The same result was observed in HeLa cells transfected with eGFP-tagged isoforms (Figure 2D). We further analysed the colocalization of the ISO1 and ISO2 isoforms by co-transfecting HeLa cells with ISO1-mCherry and ISO2-eGFP (Figure 2E, upper panel) and observed a clear colocalization of both proteins in mitochondria, with the ISO2 signal also present in the cytosol. On the contrary, when the cells were co-transfected with ISO1-mCherry and ISO3-eGFP, colocalization was absent (Figure 2E, lower panel). Collectively, this data indicates that ISO1 and ISO2 mRNA encode for the same full-length protein, whose intracellular distribution is partially different, as ISO1 is almost exclusively mitochondrial, while ISO3 mRNA encodes for the form of Lonp1 is mainly cytosolic but not mitochondrial. 

### 3.2. Overexpression of Lonp1 Isoform-1, and Isoform-2 Strongly Increased Mitochondrial Respiration

In order to understand the impact of the expression of different Lonp1 isoforms on mitochondrial functionality, we analysed the OCR, mitochondrial network, and mitochondrial dynamics. 

SW620 cells overexpressing ISO1 showed a marked increase in OCR in basal conditions when compared to wild-type cells; ISO2 showed a similar pattern, while ISO3 determined a lower increase in basal oxygen consumption. The same pattern could be observed for ATP production, while the maximal respiration capacity was markedly increased in cells overexpressing ISO1 and, to a later extent, in cells overexpressing ISO2. Cells overexpressing ISO3 did not show a significant increase in maximal respiration. Accordingly, the spare respiratory capacity was significantly increased only in ISO1 cells (Figure 3A). Moreover, in SW620 cells overexpressing ISO2, we observed an increase in ECAR, confirmed by the Western blot analysis of Lactate Dehydrogenase A (LDHA) and Glucose transporter 1 (GLUT1) (Figure 3B). Taking this together, these data suggest the higher glycolytic activity of cells overexpressing ISO2.

We hypothesized that the effect on OCR could be due to the altered morphology and functions of mitochondria that impact the efficiency of the respiratory chain. Thus, we examined the mitochondrial number and shape in the cells overexpressing the three isoforms of Lonp1 in comparison to parental cells. The number of mitochondria and the total volume were not significantly affected (Figure 3C), and the quantitative analysis of the mitochondrial mass using Mitotracker Red did not evidence any significant change among the three isoforms (not shown). In line with this observation, no significant variation in the expression of proteins involved in mitochondrial fission and fusion (namely, Opa1, Mfn1, Mfn2, and Drp1) was observed (Appendix A). Conversely, all three isoforms determined an increase in mitochondrial anion superoxide, with a higher effect of ISO3 (Figure 3D). 

As Lonp1 binds mtDNA and is necessary for its maintenance, we quantified mtDNA copies/cell in cells overexpressing Lonp1 (Figure 3E). mtDNA levels showed a high degree of variability but were not significantly affected by the overexpression of any of the isoforms.

### 3.3. Lonp1 Overexpression Increase Anchorage Independent Cell Growth, Regardless the Isoform Considered

As previously shown, the expression of Lonp1 isoforms is highly variable in cancer cells of different origins, and this could suggest a different biological role for Lonp1 isoforms in cancer development and progression.

Thus, in order to investigate if and how Lonp1 isoforms impact cancer progression, we performed a soft agar assay (Figure 4A) to highlight the possible differences in cell colony formation. While no difference in growth rate could be observed between the parental cells and cells overexpressing the different isoforms of Lonp1, cells overexpressing ISO1, 2, and 3 show a significant increment in the number of colonies (Figure 4A, right panel), while the colony size is smaller than in the control (Figure 4A, left panel). 

It was previously shown that enhanced mtROS caused by Lonp1 could activate p38 and ERK pathways, so increasing cell proliferation. However, in our model, no activation of Akt, ERK, MEK, or p38 could be observed when ISO1, ISO2, or ISO3 were overexpressed (Figure 4B,C), despite the higher levels of ROS produced. 

Since we found significant differences in the colony formation between the cells overexpressing Lonp1 isoforms and normal cells, we hypothesized the possible different roles of Lonp1 isoforms in EMT, a known process involved in cancer progression. As shown in Figure 4C, we found a significant increment in E-cadherin levels in the cells overexpressing ISO1 and 2, while not with ISO3. Furthermore, there is a slight, even if not significant, increase in N-cadherin levels when the cells overexpressed ISO1 and ISO2, while there is a decrease in Slug levels in ISO1 cells and a slight, not significant increment in Snail when ISO3 is overexpressed.

## 4. Discussion

In this study, we showed that three splicing variants of Lonp1 can be detected in human cells, which encode for three isoforms differing for their N-terminus sequence. These three isoforms display different subcellular distributions, and their expression varies significantly among cell types and between normal and transformed cells. ISO2 is expressed at low levels in the primary, non-cancerous cells we tested, while it is expressed at higher levels (representing up to 40% of total transcripts) in different cancer cell lines. More importantly, is the form whose expression increases more in cancer tissues in comparison to healthy tissue. This observation has important consequences in the evaluation of data concerning Lonp1 expression in cancer previously reported. Several authors, including us, have shown an increase in Lonp1 expression in different cancer types [21,29,30]. However, these data clearly indicate that only one of the isoforms increases, and the up-regulated isoform changes, depending on the type of cancer. It is also theoretically possible that the upregulation of an isoform could be masked, at the RNA level, by the contemporary downregulation of another isoform, with different functions. Therefore, in future studies, it will be important to clearly identify which isoform is analysed/observed when correlations between Lonp1 levels and clinical data are reported.

From the functional point of view, ISO1 and ISO2 transcripts are translated into the same full-length protein but with different subcellular localization. When overexpressed, ISO1 is almost exclusively mitochondrial, while ISO2 is mainly mitochondrial but also present in the cytosol. As we previously observed, Lonp1 can be present at the MAM, and we can hypothesize that ISO2 is the transcript that encodes the protein that localizes in this cell compartment. This can explain in an easier manner how a protein known to be mitochondrial can move to an extramitochondrial compartment. We have also observed that Lonp1 can be present in the nucleus, where it can interact with the heat shock factor-1 (HSF-1) and modulate HS response. The nuclear localization sequence that we identified in Lonp1 is preserved in all the isoforms [26], and this agrees with the observation that ISO1, ISO2, and ISO3 can be detected in the nucleus after HS. 

Contrary to the other isoforms, ISO3 is shorter and totally deprived of MTS. This agrees with its localization, which appears totally extramitochondrial. ISO3 also lacks the initial 196 amino acids, forming a part of the Lonp1 N-domain needed for substrate recognition and oligomerization. It has been previously shown that the lack of the first 270 aa impairs the proteolytic activity of Lonp1 [31], suggesting that a reduction in the proteolytic activity of ISO3 could occur. However, ISO3 is 74 aa longer than that of the form analysed by Kereiche et al. [31]. Further studies on the purified ISO3 could clarify to what extent its proteolytic activity is preserved. As ISO1/ISO3 colocalization is negligible, it is likely that their coexpression, normally observed in cells, does not lead to the formation of hetero-oligomers with reduced enzymatic activity. 

The overexpression of Lonp1 isoforms has different effects on mitochondria. Although the overexpression of ISO1 and ISO2 has a similar impact on respiration, some subtle differences emerged. ISO1 overexpression determined higher maximal respiration and, on top of that, a significantly higher SRC. Mitochondrial SRC is a crucial aspect of mitochondrial function, as it increases resistance to stress and better maintenance of cellular functions when the energy demand increases, as in cancer cells [32]. SRC depends mainly on the integrity of the electron transport chain and on mitochondrial homeostasis. Since Lonp1 in the mitochondrial matrix is involved in the proper folding of ETC proteins, in their maintenance of ETC complexes (particularly complex I), and in mitophagy regulation, it is likely that higher levels of ISO1, which is totally imported in the mitochondrial matrix, rather than ISO2, allows cells to better maintain mitochondrial homeostasis. This capability to increase OCR appeared unrelated to changes in the mitochondrial network, as the total volume of mitochondria, their number, and shape were unaffected by Lonp1, whatever the isoform considered, and as to mitochondrial dynamics, no relevant changes have been observed in proteins regulating fusion or fission. 

Although ISO1 cells have higher respiration reserves, ISO2 cells have an ECAR similar to ISO1, indicating that both these isoforms (and, to a lesser extent, ISO3) can induce an increase in glycolysis. The ECAR, linked to the glycolytic flux, resulted in significantly lower ISO1 cells when compared to ISO2 in basal conditions. Conversely, ISO1 and ISO2 have overlapping ECAR levels under conditions maximizing the glycolytic flux (i.e., in the presence of oligomycin), thereby suggesting a better capability to undergo a metabolic switch to the glycolysis to compensate for a respiratory deficit. We also found that ISO2 cells had higher levels of LDHA and GLUT1 than the parental cells. Previous studies–including ours—reported that Lonp1 overexpression determined a metabolic reprogramming, with a glycolytic switch associated with the maintenance of the same respiration capacity [33], and we previously showed that Lonp1 could induce LDHA and G6PD expression in SW620 cells. Our data indirectly suggest that when ISO2 is expressed (that is, Lonp1 is present in the mitochondria and outside the mitochondria), cells have a better capability to increase glycolysis. We can speculate that the ratio between ISO1 and ISO2 mRNA determines the relative amount of Lonp1 that enters the mitochondria and the final effect of Lonp1 on mitochondrial respiration and glycolysis.

In agreement with previous observations, Lonp1 overexpression determined an increase in mt ROS, which was more evident in cells overexpressing ISO2 and ISO3. A previous report connected a higher production of ROS at complex I, caused by LonP1 overexpression, to the activation of the ERK and p38/JNK pathway, which in turn determined a higher proliferation rate and EMT [23]. Our data did not confirm this link, as we did not observe any relevant activation of these pathways nor a correlation with the relative levels of ROS. The fact that mtROS levels were higher when ISO2 and ISO3 were overexpressed clearly indicates that mtROS production in Lonp1-overexpressing cells is not mediated exclusively by the upregulation of NDUFS8 at complex I, as proposed by Cheng et al. [23]. 

Cells overexpressing ISO1, ISO2, or ISO3 do not differ in terms of their proliferation rate, and apparently, Lonp1 overexpression did not confer any proliferative advantage in comparison to the parental cells, independently from the isoform expressed. Conversely, Lonp1 overexpression dramatically increased the capability of the cells to form colonies in an anchorage-independent environment. This effect is slightly higher—although not reaching statistical significance—in ISO3 cells. This is in line with previous observations that proved a higher metastatic potential of cancer cells expressing high levels of Lonp1 in vivo and with clinical data that, at least in some types of cancers, associated higher Lonp1 levels with a poorer prognosis. As it is also observed in ISO3 cells, this effect cannot be attributed to the direct effect on mitochondrial homeostasis or to changes in respiration. Although not formally proven, we could suppose that this effect is likely mediated by modulating the expression of EMT markers in ISO1 and ISO2 cells that present similar behaviour. However, it is interesting to note that in ISO1 cells, there is a clear downregulation of SLUG. As the EMT transcription factors can exhibit the consistent activation of glycolysis, this could be one of the reasons that explain the more marked increase in ECAR observed in ISO2 cells. 

The biological significance of ISO3 remains to be determined. The observation that its expression is barely detectable in many cell types strongly suggests that its function(s) is dispensable for cell homeostasis. Experiments are ongoing to determine its proteolytic activity in vitro.

In conclusion, our study revealed the presence of three isoforms of Lonp1 in the cells, with a different intracellular distribution pattern, and a complex change in their relative expression of these isoforms in different cancer types, rather than a simple upregulation of the gene, as previously described.

## Figures and Tables

**Figure 1 cells-11-03940-f001:**
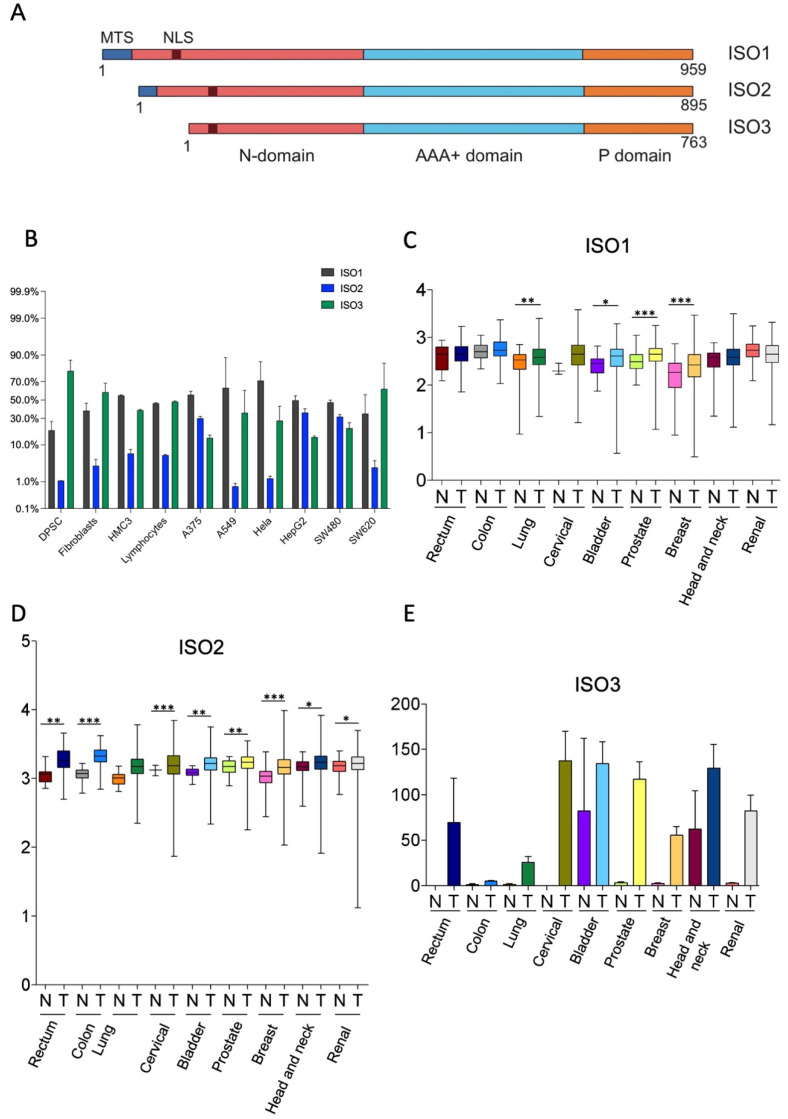
Human cells express three isoforms of Lonp1. (**A**). Diagram depicting the location of mitochondrial targeting sequence (MTS), of the nuclear localization signal (NLS), and of the N, AAA+ and P domains of three isoforms of human Lonp1. The length of the three isoforms is indicated. (**B**). Histogram showing the percentage of Lonp1 isoform-1 (ISO1), Lonp1 isoform-2 (ISO2), and Lonp1 isoform-3 (ISO3) mRNA expression in primary cells (DPSC, dental pulp stem cell; Fibroblasts; HMC3, human microglial cell; Lymphocyte) and cancer cell lines (A375, melanoma; A549, lung cancer; Hela, human cervix epitheloid carcinoma; HepG2, hepatocellular carcinoma; SW480, non-metastatic colon adenocarcinoma; SW620, metastatic colon adenocarcinoma) of different origins. Data represent the mean ± SD. (**C–E**). Relative expression levels of Lonp1 ISO1, ISO2, and ISO3 mRNA in the indicated primary solid tumours (T) and the normal (N) counterparts, as obtained from TGCA Splicing Variant database (TSVdb). Rectum: rectum adenocarcinoma; Colon: colon adenocarcinoma; Lung: lung adenocarcinoma; Cervical: cervical squamous cell carcinoma and endocervical carcinoma; Bladder: bladder urothelial carcinoma; Prostate: prostate adenocarcinoma; Breast: breast invasive carcinoma; Head and neck: head and neck squamous cell carcinoma; Renal: renal clear cell carcinoma. Data are in log scale; mean, 24 and 75 percentile, max and min values are shown. Data for ISO3 are in linear scale and expressed as mean ± SD * = *p* < 0.05; ** = *p* < 0.01; *** = *p* < 0.0001.

**Figure 2 cells-11-03940-f002:**
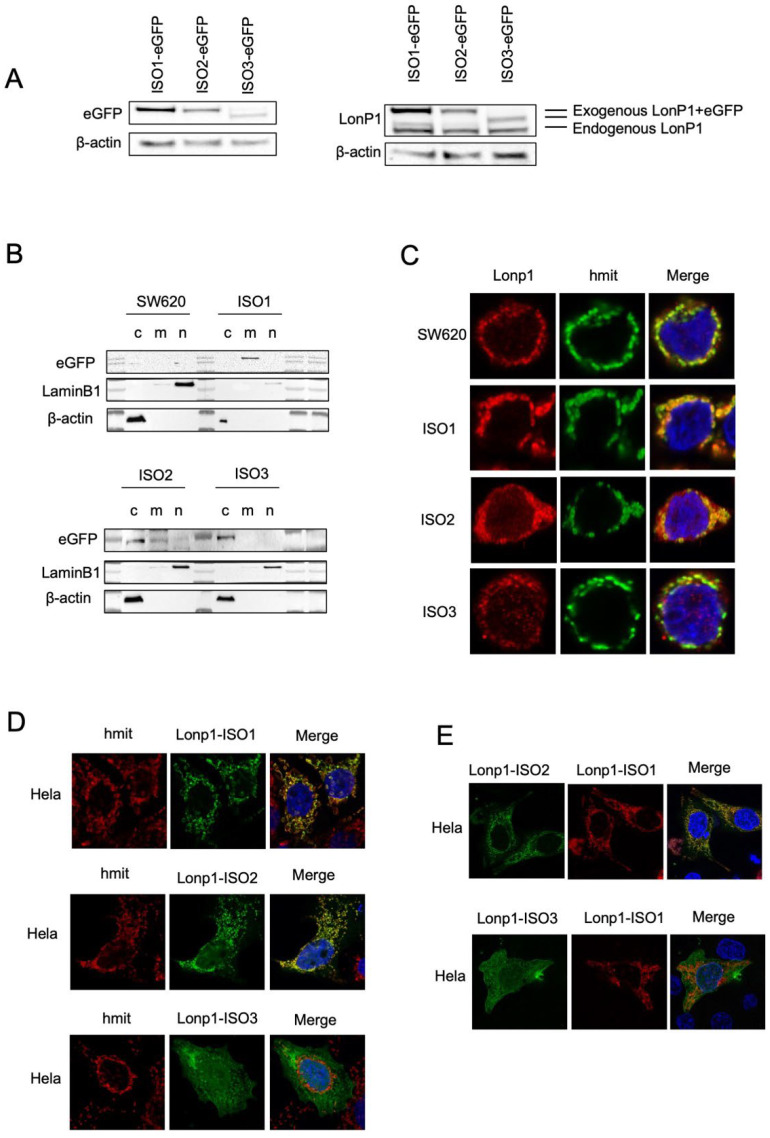
Cellular localization and distribution of three isoforms of Lonp1. (**A**). Immunoblot detecting Lonp1 using anti eGFP (left panel) and anti Lonp1 (right panel) Abs in cells transfected with Lonp1 ISO-eGFP, ISO2-eGFP, or ISO3-eGFP. The position of the bands corresponding to the exogenous, eGFP-tagged form and the endogenous form of Lonp1 is indicated. (**B**). Representative immunoblots showing Lonp1 expression in cytosolic (C), mitochondrial (M), and nuclear (N) fractions in SW620 cells transfected with constructs bearing Lonp1 isoforms tagged at C-term with enhanced green fluorescent protein (eGFP). Lonp1 has been detected using anti-eGFP antibody. (**C**). Representative confocal microscopy images of SW620 cells transducted with three different constructs bearing Lonp1 isoforms tagged at C-term with enhanced green fluorescent protein (eGFP), namely Lonp1-ISO1, Lonp1-ISO2, or Lonp1-ISO3. Mitochondria were stained with anti-hMit and nuclei were counterstained with DAPI. (**D**). Representative confocal microscopy images of HeLa cells transfected with Lonp1-ISO1, Lonp1-ISO2, or Lonp1-ISO3. Mitochondria were stained with anti-hMit and nuclei were counterstained with DAPI. (**E**). Representative confocal microscopy images of HeLa co-transfected with constructs bearing Lonp1 isoforms tagged with enhanced green fluorescent protein (eGFP, in green) or mCherry (in red). Nuclei were counterstained with DAPI.

**Figure 3 cells-11-03940-f003:**
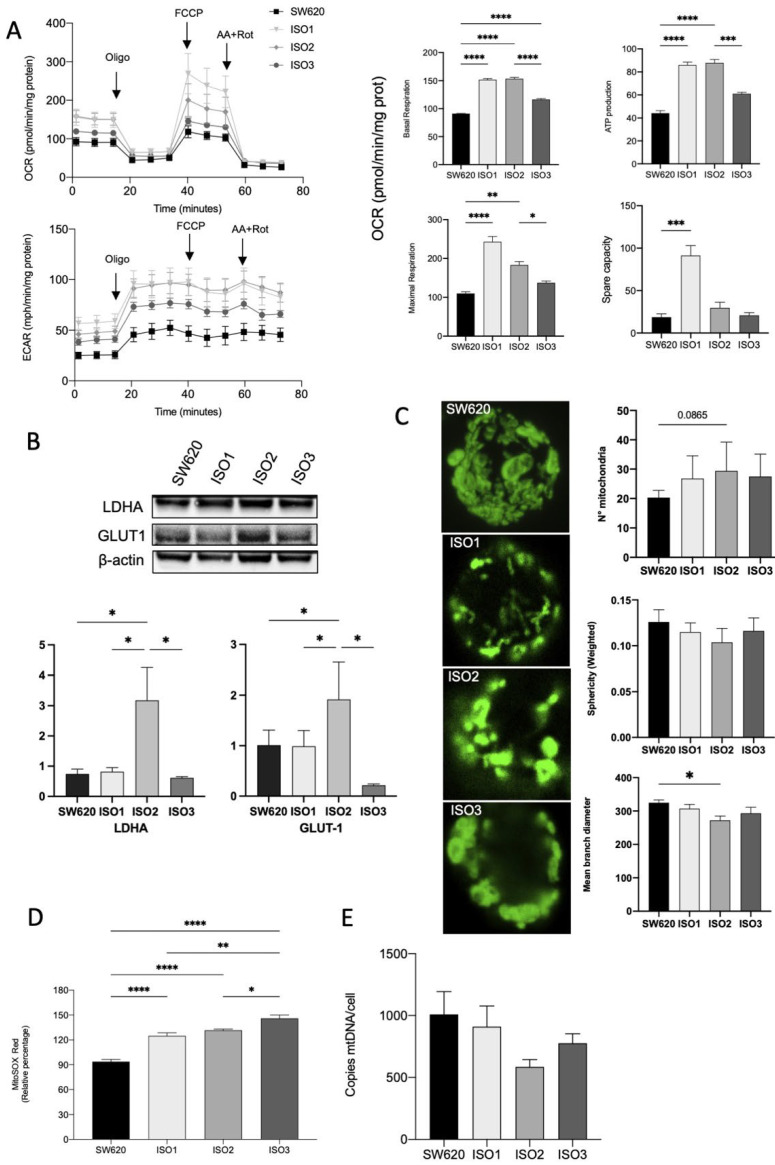
Effects of Lonp1 isoform-1, isoform-2, and isoform-3 overexpression on mitochondrial functions. (**A**). Seahorse analysis of oxygen consumption rate (OCR) and extracellular acidification rate (ECAR) in SW620 and cells overexpressing Lonp1-ISO1, Lonp1-ISO2, or Lonp1-ISO3 (on the left). Histograms showing single parameters of OCR, basal respiration, ATP production, maximal respiration, and spare capacity in the same cell lines (on the right). Data are reported as mean ± SD of three independent experiments. (**B**). Representative immunoblots showing LDHA and GLUT1 expression in SW620 and cells overexpressing Lonp1 isoform-1, isoform-2, and isoform-3 (upper panel). Histograms showing the relative expression of LDHA and GLUT1 in the same cell lines (lower panel). Data are reported as mean ± SD of three independent experiments. (**C**). Representative confocal microscopy images of mitochondria in SW620 and cells overexpressing Lonp1-ISO1, Lonp1-ISO2, or Lonp1-ISO3 (on the left). Mitochondria were stained with anti-hMit. Histograms showing the analysis of total number of mitochondria, sphericity, and mean branch diameter in SW620 and cells overexpressing Lonp1-ISO1, Lonp1-ISO2, or Lonp1-ISO3 calculated by FIJI (ImageJ) (on the right). Data are reported as mean ± SD of ten independent experiments. (**D**). Histogram showing ROS relative percentage in SW620 and overexpressing Lonp1-ISO1, Lonp1-ISO2, or Lonp1-ISO3. Data are reported as mean ± SD of three independent experiments. (**E**) Histogram showing the number of copies of mtDNA per cell is SW620 cells overexpressing Lonp1-ISO1, Lonp1-ISO2, or Lonp1-ISO3. Data are reported as mean ± SD of three independent experiments. * = *p* < 0.05; ** = *p* < 0.01; *** = *p* < 0.001; **** = *p* < 0.00001.

**Figure 4 cells-11-03940-f004:**
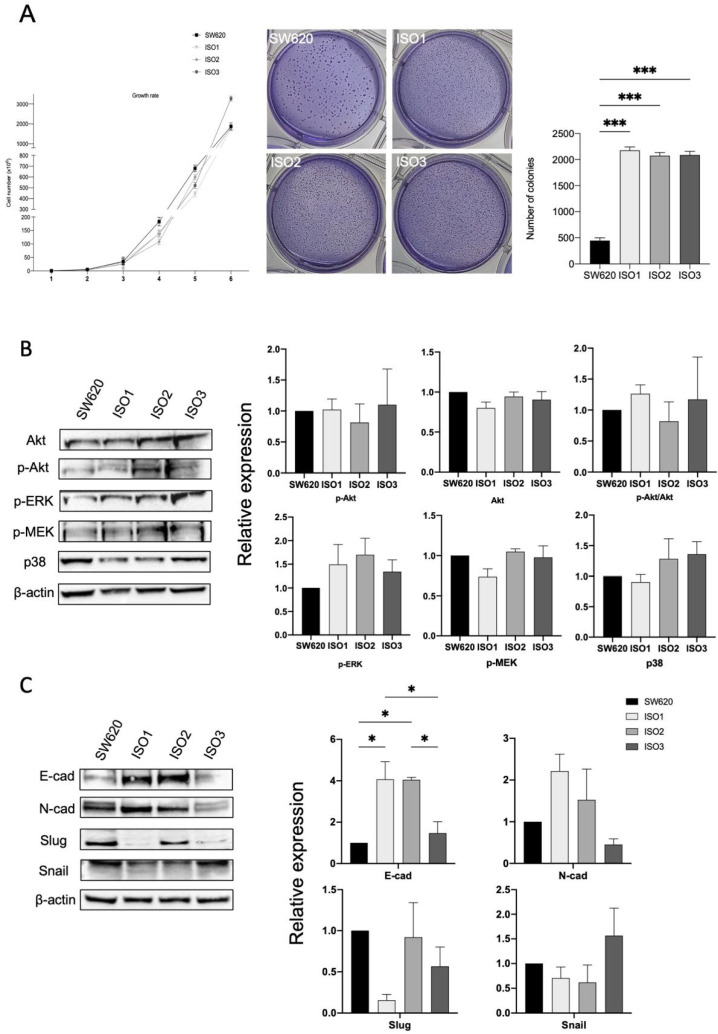
Lonp1 isoforms can affect cancer development and progression. (**A**). Graph showing the growth rate of SW620 and cells overexpressing Lonp1 isoform-1, isoform-2, and isoform-3 (on the left). Soft agar colony formation assay of SW620 and cells overexpressing Lonp1 isoform-1, isoform-2, and isoform-3 (in the middle). Histogram showing the number of colonies, from soft agar colony formation assay, of SW620 and cells overexpressing Lonp1 isoform-1, isoform-2, and isoform-3 (on the right). Data are reported as mean ± SD of two independent experiments. (**B**). Representative immunoblots showing Akt, P-Akt, P-ERK, P-MEK, and p38 expression in SW620 and cells overexpressing Lonp1 isoform-1, isoform-2, and isoform-3 (on the left). Histograms showing the relative expression of Akt, P-Akt, P-Akt/Akt, P-ERK, P-MEK, and p38 in the same cell lines (on the right). Data are reported as mean ± SD of three independent experiments. (**C**). Representative immunoblots showing E-cad, N-cad, Slug, and Snail expression in SW620 and cells overexpressing Lonp1 isoform-1, isoform-2, and isoform-3 (on the left). Histograms showing the relative expression of E-cad, N-cad, Slug, and Snail in the same cell lines (on the right). Data are reported as mean ± SD of three independent experiments. * = *p* < 0.05; *** = *p* < 0.0001.

## Data Availability

The data presented in this study are available on request from the corresponding author.

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
