# Peer review of "Differential Expression of Lonp1 Isoforms in Cancer Cells"

_cells, 2022, doi:10.3390/cells11233940_

Round 1
Reviewer 1 Report
The article "Differential expression of Lonp1 isoforms in cancer cells" reports data on the possible consequences of the expression of alternatively spliced isoforms of Lonp1. This work is however mostly descriptive and does not bring any clear insights in terms of mechanistic. Moreover, most of the cell biology experiments results from the overexpression of the 3 isoforms. This makes conclusion in a way quite hazardous since artefact might arise from the high levels of expression of the 3 proteins. To me, the article should be reinforced by working on the endogenous forms. The inhibition of the expression of the different isoforms should also help confirming or not the putative roles of the three isoforms in cancer cells. Some additional comments and specific concerns/questions are outlined below.
Main comments:
(1) The exact nature of ISO2 and ISO3 should be clarified. As such, several sentences are very confusing. For example, line 242, the authors wrote “at least two other isoforms are present in humans, resulting from alternative splicing of exon 1”. However, I do not understand how the alternative splicing of the same exon might lead to ISO2 to ISO3. Similarly, line 307, the authors wrote “ISO1 and ISO2 mRNA encode for the same full-length protein”. A similar sentence can be read on line 422. What does that mean? This is not in agreement with the drawing on figure 1 or the sentence line 408. If this is the same protein, why does their localization differ? To me, a clear drawing should be made on figure 1. It should include the whole gene organization (exons + introns) to identify which exons are alternatively spliced. Then, primers used for quantifying the different Lonp1 isoforms should be indicated. And more importantly, the constructs should be verified to make sure this is truly ISO1 or ISO2 we observe.
(2) Lines 182-184: the exact procedure should be clarified. As such, it is very unclear to me how the authors distinguished the 3 alternatively spliced isoforms. Moreover, how do the authors normalize the levels of Lonp1 mRNA?
(3) On figure 1, the authors show that ISO3 mRNA is expressed at high level in SW620 cells. On the contrary, the authors show that the endogenous form of ISO3 protein is not detectable in SW620 cells. Is that true for all the cell lines tested? This is important since the authors mostly focus on the fact that Lonp1 is overexpressed in cancer cells. However, this overexpression at the protein level might truly only concern ISO1 and ISO2, not ISO3. If this is the case, most of the data obtained here where ISO3 is overexpressed would be irrelevant.
(4) Figure 2A, right panel: what is the upper band seen in the ISO3-eGFP lane? It has the same size than ISO1 and ISO2-eGFP.
(5) Figure 4A and 4C: the authors link EMT marker expression changes and soft agar growth for ISO1 and ISO2. However, this does not apply to ISO3. Therefore, how to make sure there are any links between the 2 observations for ISO1 and ISO2?
(6) Line 410: the authors mentioned that “ISO2 is the form expressed at higher transcriptional levels in many cell types”. This is clearly not true for most of the cell lines tested in this paper (see figure 1A). This issue should be discussed. Moreover, we can see that we have a clear disconnection between mRNA and protein expression levels for ISO3. This can also be true for ISO2. Therefore, this makes the assumption made by the authors very speculative.
(7) Line 419: the authors indicate that “it will be important to clearly identify which isoform is analysed/observed when correlations between Lonp1 levels and clinical data are reported”. To me, this is an important issue since most of the paper relies on this hypothesis. Therefore, this should be done now, not in the future.
(8) Line 464: the authors state that “Our data indirectly suggest that when ISO2 is expressed (that is Lonp1 is present in the mitochondria and outside mitochondria), cells have a better capability to increase glycolysis”. Please clarify and/or expand. Moreover, to me, on Figure 3A lower panel, this is when ISO1 is overexpressed that glycolysis reach its higher level.
(9) Lines 492-494: this part of the discussion is highly speculative and requires more mechanistic studies. Additional experiments should be done to bring clear insights to the observation made here.
(10) Lines 499-502: the authors conclude: “In conclusion, our study revealed the presence of three isoforms of Lonp1 in the cells, with a different intracellular distribution pattern, and different impact on cell metabolism, and a complex change in the relative expression of these isoforms in different cancer types, rather than a simple upregulation, as previously described”. To me, we cannot draw such conclusions from the data presented here since all data refer to the overexpression of exogenous forms of Lonp1. Such hypotheses should be confirmed by experiments done on endogenous protein isoforms. For example, can we correlate any metabolic changes to the endogenous ISO1, ISO2 or ISO3 levels of expression measured in different cell lines? As such, it is quite unlikely to me.
Minor comments:
(1) line 210: the meaning of “spare respiration” should be indicated here not in the discussion (line 446).
(2) lines 261-263: the authors write “the expression of ISO3 was detectable in some normal cell types, such as rectum and cervical epithelium, and did not show any significant changes in any types of cancer considered”. If “T” means “tumor sample”, the meaning of the sentence is wrong since ISO3 mRNA are exclusively expressed in T, not in N. Please clarify or correct.
(3) line 303: “Figure 2D” should be corrected to “Figure 2E lower panel”.
(4) line 306: “Figure 2E” should be corrected to “Figure 2E upper panel”.
(5) Figure 2C: what is shown on the upper lane “SW620”? Is that the endogenous form?
(6) line 346: I could not find supplementary Figure 1.
(7) Figure 3A, 3B and 3C: Symbols indicating statistical significance (“*”) are missing.
(8) Figure 3A, left panels: it would be helpful for the reader to indicate when the different drugs are added to your sample. Then, it is easier to figure out what is effectively measured.
(9) line 383: the following sentence “Lonp1 overexpression facilitates cell invasion/migration” should be deleted. There are no data on invasion/migration here.
(10) Figure 4A and 4C: Symbols indicating statistical significance (“*”) are missing.
Reviewer 2 Report
In the manuscript from Giada Zanini et al. the authors combined different approaches to investigate the pattern of expression and a possible fuctional role of three different isoforms of the mitochondrial protease Lonp1 in normal and tumor cells. The methods used in the study and the results are well described and discussed. The authors show, by reporter gene expression, confocal microscopy and western blot analyses that the three isoforms have different cellular localization and affect differently cell metabolism in different cancer types.
Minor suggestion:
In the Abstract row 26, “ISO” should be “ISO1”; In the Introduction row 63, “promotes” should be deleted;
In figure 1B the fonts should be enlarged to make them more legible and the figure should be in color like the others;
In figure 3 and 4 are missing the asterisks to indicate the p-value significance as explained in the legend to the figures.
Round 2
Reviewer 1 Report
Dear authors, thank you for your detailed response and the clarification made. To me, the key point in my new decision is the data your present on patients' samples and the putative prognostic value of looking at the expression levels of the different isoforms. Then, it becomes acceptable to me of using overexpressed forms of the three different isoforms to better understand their role and localization. It also pave the way for setting up "home-made" clinical studies to confirm the impact of Lonp1 isoforms expression on cancer patients outcome.